# 2D Sonic Acoustic Barrier Composed of Multiple-Row Cylindrical Scatterers: Analysis with 1:10 Scaled Wooden Models

Antonella Bevilacqua [1],*, Gino Iannace [2],*, Ilaria Lombardi [3] and Amelia Trematerra [2]

1 Department of Industrial Engineering, University of Parma, Area delle Scienze, 43126 Parma, Italy
2 Department of Architecture and Industrial Design, University of Campania "Vanvitelli", Via Michelangelo, 81031 Aversa, Italy; amelia.trematerra@unicampania.it
3 Department of Industrial Engineering, University of Campania "Vanvitelli", Via Michelangelo, 81031 Aversa, Italy; ilaria.lombardi@unicampania.it
* Correspondence: antonella.bevilacqua@unipr.it (A.B.); gino.iannace@unicampania.it (G.I.)

**Abstract:** Theory regarding metamaterials was developed in the 1960s, aiming to control the propagation of electromagnetic waves. Under this scope, research has been focused on the realization of materials having specific characteristics to be invisible to the electromagnetic and optics fields. These principles have been expanded only recently to the acoustic sector, with metamaterials capable of controlling the sound propagation due to the interference effect between the soundwaves and the periodic structural elements composing the system. This paper deals with sound attenuation and analyzes a metamaterial acoustic barrier characterized by multiple rows in different configurations. The variety of configurations depends on different diameters of the wooden scatterers (i.e., 9 mm and 15 mm) and the distance between the sound source and the closest edge of the barrier (i.e., 400 mm and 800 mm). Despite having the same height (i.e., 300 mm) of a scaled model, the combination of different diameters in creating an acoustic barrier highlights an increase of the overall Insertion Loss (IL) and a broadened (instead of sharp) sound attenuation of the band gap, captured between 4 kHz and 12.5 kHz.

**Keywords:** metamaterials; modular bars; insertion loss; scaled model; noise attenuation; cylindrical bars; periodic structure; 2D sonic crystals

## 1. Introduction

Metamaterials are artificial materials composed of periodic cellular structures not existing in nature [1]. This innovation represents a new benchmark in the field of applied acoustics, since the control of the sound propagation is strictly dependent on the interaction between the incident sound waves and the geometry of the external surface [2]. The presence of modular elements composing the periodic structures is favorable in creating such sound attenuation through the destructive interference of the sound waves when the latest ones penetrate the gaps of the whole structure [3]. The scattering effects generated by regular and periodic geometries were initially studied in the field of electromagnetism. This concept follows the regime of Bragg's law, such that the band gap frequency ($f_{BG}$) depends on the incident angle of the wave ($\phi$), the lattice constant ($\alpha$), and the sound speed in the medium ($c$) [4], as summarized in Equation (1).

$$f_{BG} = \frac{c}{2\alpha \, \sin(\phi)} \tag{1}$$

On this basis, when a plane wave affects a structure composed of elements arranged with a regular geometry, the distance between the wavefronts can be described as $a \, \sin(\phi)$, where ($\phi$) is the angle of incidence of the plane wave, $a$ is the distance between two consecutive rows of scatterers, and $\lambda$ is the wavelength of the incident wave. The condition

for the destructive interference is that $\lambda/2 = a \sin(\phi)$. At normal incidence, this implies that the stop band or the Bragg's pass band frequency is equal to $f_{BG} = \frac{c}{2a}$. Figure 1 shows the scattering of a plane sound wave when hitting a barrier composed of two rows of scatterers.

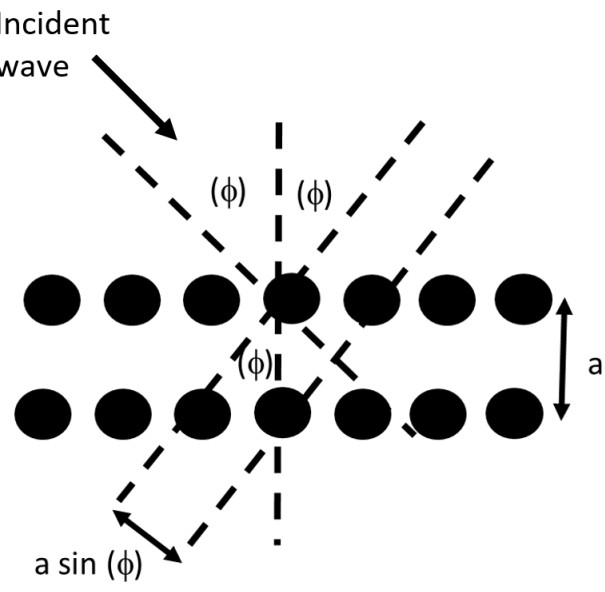

**Figure 1.** Scattering phenomenon of plane sound waves across a two-row array of cylindrical scatterers.

By applying these theories to acoustics, development of the sonic crystals took place, since the metamaterials are considered solid materials, subject to interaction with a dynamic fluid (i.e., air, water). Under these conditions, sound attenuation can be achieved at any frequency, despite some traditional sound-absorbing materials that are not suitable for this scope [5,6].

According to V. Veselago and J. Pendry [7], specific physical properties related to the elastic modulus, mass density, and refractive index occur to create the destructive interference and hence the band gap. The IL is strictly related to the density of the material used for the scatterers, to the ratio of the volume occupied by the scatterers in a determined volume of the crystal that is called the filling factor (*ff*), and to the lattice design, which can assume different shapes [8].

Generally, the band gap attenuation that is aimed to be achieved for road traffic noise, characterized by a medium frequency wavelength, depends on the cylinder's radius rather than on the space between scatterers, while the amplitude of attenuation depends on the filling factor [9]. To obtain an effective frequency-selection attenuation, the modularity or the periodicity of the scatterers' arrangement shall be in place along with the distance between source and barrier.

The baseline principle of creating an acoustic barrier, composed of a certain number of cylindrical scatterers periodically installed, represents the phenomenon of diffraction that redirects the streaming of the soundwaves across the voids, as shown in Figure 2.

This technique has also been used by artists (e.g., Eusebio Sempere), who arranged the installation of artworks based on specific characteristics, listed as follows:

- Selection of a geometric shape for the scatterers' configuration,
- Determination of the distance between elements,
- Height of the modular scatterers (constant/irregular),
- Selection of scatterers' diameter.

The periodic-structure artworks represent a source of inspiration for sound crystal research, which tries to reduce or mitigate road traffic noise pollution in accordance with the divergence phenomenon. Research studies on this topic highlight how the gap band can be shifted across the spectrum by undertaking tests on the same structure but changing the

variables as per the above discussion [10]. This outcome is mainly due to the constructive or destructive interference that varies the amplitude of the emitted soundwaves [11–13].

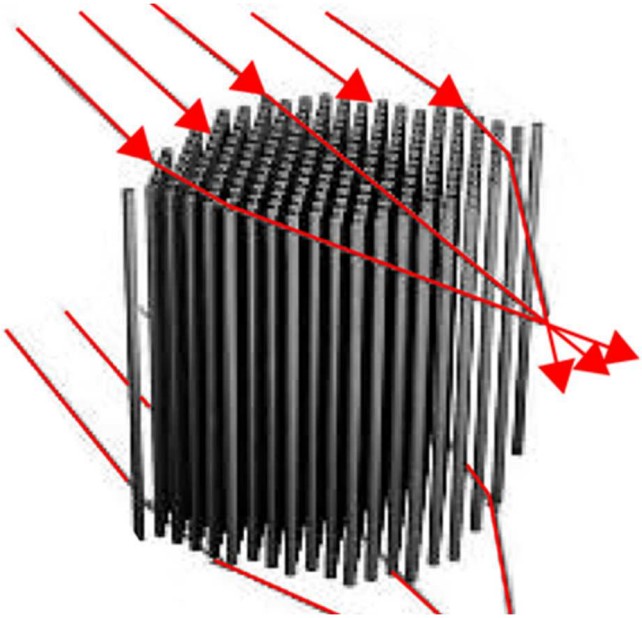

**Figure 2.** Scheme of sound control through the structure of a sonic crystal barrier.

Among the different categories and their applicability, this paper deals with 2D sonic crystals only, composed of regular arrays (or lattice design) of solid wooden cylinders, coupled for two different sizes of diameters in multiple rows, and tested with a scaled model with the purpose of implementing the research studies on barriers for road traffic noise mitigation. Previous research studies [14] have been carried out on real, scaled models, realized by using wooden bars of a 300 mm height. The contribution of this paper is the use of different sizes of diameters, equal to 15 mm and 9 mm. The overall dimensions have been scaled by a proportion of 1:10 compared to the effective size that should be assumed. Acoustic tests have been carried out to investigate the attenuation efficiency of different configurations of multiple-row barriers by spacing the scatterers at a constant distance.

The outcomes highlight that the sound attenuation is directly proportional to the increased number of rows and is more effective by coupling two sizes of diameter. Results are reported in terms of an insertion loss (*IL*) that mirrors the attenuation obtained by testing multiple-row barriers installed between the source and the receiver.

Traditionally, acoustic barriers have been considered solid screens by having the function of shielding any noise source from sensitive receptors. Although the aesthetic design of the barriers has been improved in the latest decades, also given the availability of innovative and flexible materials that enhanced the sectional shape [15], the visual impact always represents a concern for urban design, aggravated further by a social aspect that considers these construction elements as the ingredients for exclusion. Nowadays, the research is aligning with different applications of acoustic barriers to be aesthetically more pleasant, even along the edges of roads. In this direction, studies focused on the absorption of low-frequency noise from outdoor sources (e.g., road traffic) have been undertaken by designing trees to be planted on a regular-grid base [16]. Investigations have been carried out for the purpose of having solid and empty entities composing the metamaterial barrier, highlighting no difference between soundwaves when interacting with a solid and opaque surface [17,18]. As such, acoustic tests undertaken on green barriers, e.g., natural bars such as bamboo, have shown reliability on a certain frequency band of sound attenuation, given its nature to be tubular and empty in the core [19]. In this paper, the choice of undertaking experiments on plain wood bars is due to simulating tree trunks in real dimensions rather

than dictated by practical reasons, including the shear cutting of tubular materials that often results in cracking and makes materials not suitable to be subject to experiments. Nonetheless, both bamboo and plain wood composed of discarded chunks are eco-friendly.

This paper aims to contribute to understanding the behavior of multiple-row and coupled acoustic barriers using different applications. In this specific case, the wooden bars are organized on a surface area of 9 m$^2$ (a sample of 3 × 3 m), which highlights the strengths and weaknesses of the selected lattice design. Along with other research studies mainly focused on the road traffic abatement (having a broad band noise centered on 1 kHz) [20–22], the proposed acoustic barrier has a high absorption, in the range between 800 Hz and 1250 Hz.

The bars used for the scaled models are cylindrical plain wooden scatterers, having a density of approximately 600 Kg/m$^3$ and a modulus of elasticity (MOE) around 12 GPa, with a 300 mm height and diameters of the sectional areas equal to 15 mm and 9 mm. The bars are spaced with a dimension equal to the diameter of the bars themselves, and the model is 1.0 m long. As already discussed, a wide selection of materials can be used for creating the scatterers; the choice of wood allows compliance with the current guidelines and regulations for green energy and waste recycling. Furthermore, wood is broadly used outdoors, given its long-term resistance to meteorological conditions. Regarding the sectional area of the bars, a solid circle was preferred to any concave shape, which potentially might have openings and/or holes, because of practical reasons. The accumulation of dust is most likely to fill cavities and alter the outcomes.

## 2. Materials and Methodologies

### 2.1. Generalities and Test Conditions

The acoustic barriers herein investigated were realized in a 1:10 model [23] that was tested by considering 1/3-octave bands from 1 kHz to 15 kHz, equivalent to the range 100–1000 Hz in the case of a real scale [24–26].

Acoustic measurements were undertaken inside a semi-anechoic chamber, to be 4.40 × 4.40 × 4.50 m (L × W × H), composed of absorbing materials on walls and MDF boards on the floor and having a cut-off frequency equal to 100 Hz, according to Schroeder's theory [27]. The arrangement of placing hard floor finishes with MDF boards simulates the real configuration of a road environment, having acoustic barriers on the sides. On this basis, the acoustic measurements consider both direct and reflected soundwaves.

The results were analyzed in terms of insertion loss (*IL*), considered as the configuration difference between the sound pressure level measured without any obstacle ($L_{FF}$) and with the presence of a barrier ($L_{Bar}$) [28], as indicated in Equation (2).

$$IL = L_{FF} - L_{Bar} \text{ (dB)} \tag{2}$$

In terms of notation, Equation (2) derives from the ratio of the incident wave $p_{inc}$, which can be either plane or cylindrical, and the total field $p(r)$, as indicated in Equation (3).

$$IL = 20log_{10} \left| \frac{p_{inc}}{p(r)} \right| \text{ (dB)} \tag{3}$$

In a controlled environment like an anechoic chamber, the diffraction of the lateral edge of the barriers can be considered negligible, as well as all the other factors (e.g., air absorption, wind speed, temperature variance, etc.) that contribute to modifying the spreading of the sound waves. On this basis, the results should be considered as more optimistic than the true effects of a real environment.

By considering *Xs* and *Xr* as the distance occurring between the source or the receiver and the edge of the barrier, as shown in Figure 3, the dimensions used for the first set of measurements are the following:

- *Xs* = 800 mm;
- *Xr* = 800 mm.

For the second set of measurements, the following dimensions of *Xs* and *Xr* were used:

- *Xs* = 400 mm;
- *Xr* = 800 mm.

The third set of measurements was carried out in order to visualize the spatial distribution of the IL modeled on the receiving side.

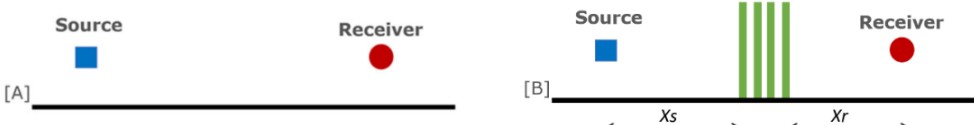

**Figure 3.** Sectional scheme of the equipment location during the measurements without (**A**) and with (**B**) the metamaterial acoustic barrier.

Given the employment of an omnidirectional sound source, instead of a linear array loudspeaker, the limits of these tests were optimized by moving the receiver across an x–y plan (without any height alteration) in order to create different directivity angles. For this scope, Figure 4 indicates in plan the disposition of the equipment during the assessment. The position of the receiver follows a regular grid having a distance between nodes equal to 200 × 200 mm. On this basis, the microphone was moved to 25 positions, while the sound source was fixed on the central axis of the barrier at an 800 mm distance. This methodology was adopted to simulate the movement of a linear source, as the road traffic would be.

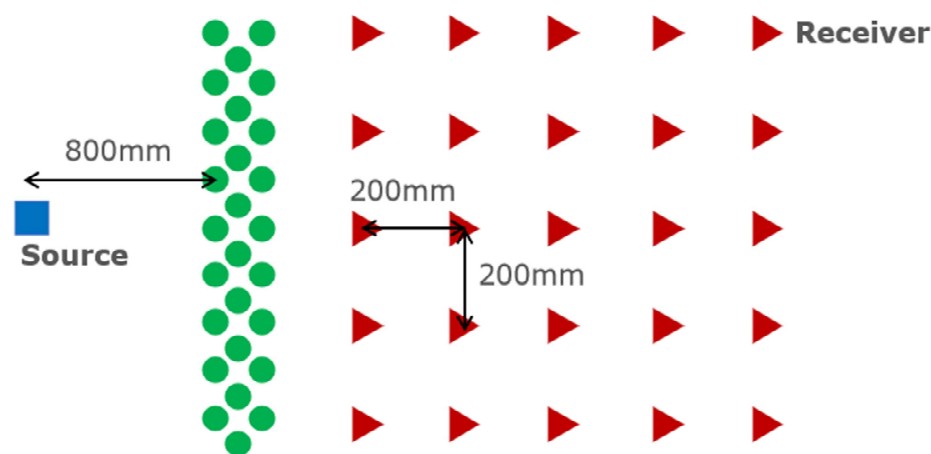

**Figure 4.** Schematic layout of the equipment location during the third set of measurements.

Where the scatterers have a diameter of 15 mm (A), the centered distance between wooden sticks is 30 mm (2A). Where the cylindrical elements have a diameter of 9 mm (B), the centered distance between wooden sticks is 18 mm (2B).

### 2.2. Acoustic Measurements

The surveys were carried out with the following equipment:

- Dome tweeter sound source (RCF TWT 50);
- Condenser microphone (Audiomatica 1/4-inch) [29];
- Audio interface (Clio CP-01);
- Computer connected to the compact audio interface.

The selection of the sound source was a flat response (under free field conditions) in the frequency range where the analysis was focused. The excitation sound signal was an Exponential Sine Sweep (ESS) having a time-invariant sound pressure level. A compact Clio CP-01 audio interface was used to feed the loudspeaker with the sound signal coming from the computer and to record the impulse response (IR) [30]. The sound source was placed at 45 mm from the finish floor, while the microphone was 80 mm high.

In accordance with Bragg's law and with the dimension of scatterers, the theoretical acoustic attenuation (or gap band frequency) occurred at 3.1 kHz.

The data acquisition of the recorded signal was based on a 65,000-point FFT and on a frequency sample rate of 48 kHz. The frequency range being considered was found to be between 1 kHz and 12.5 kHz, equivalent to 100 Hz and 1.25 kHz on a real scale. Figure 5 shows the configurations of all the measurement settings.

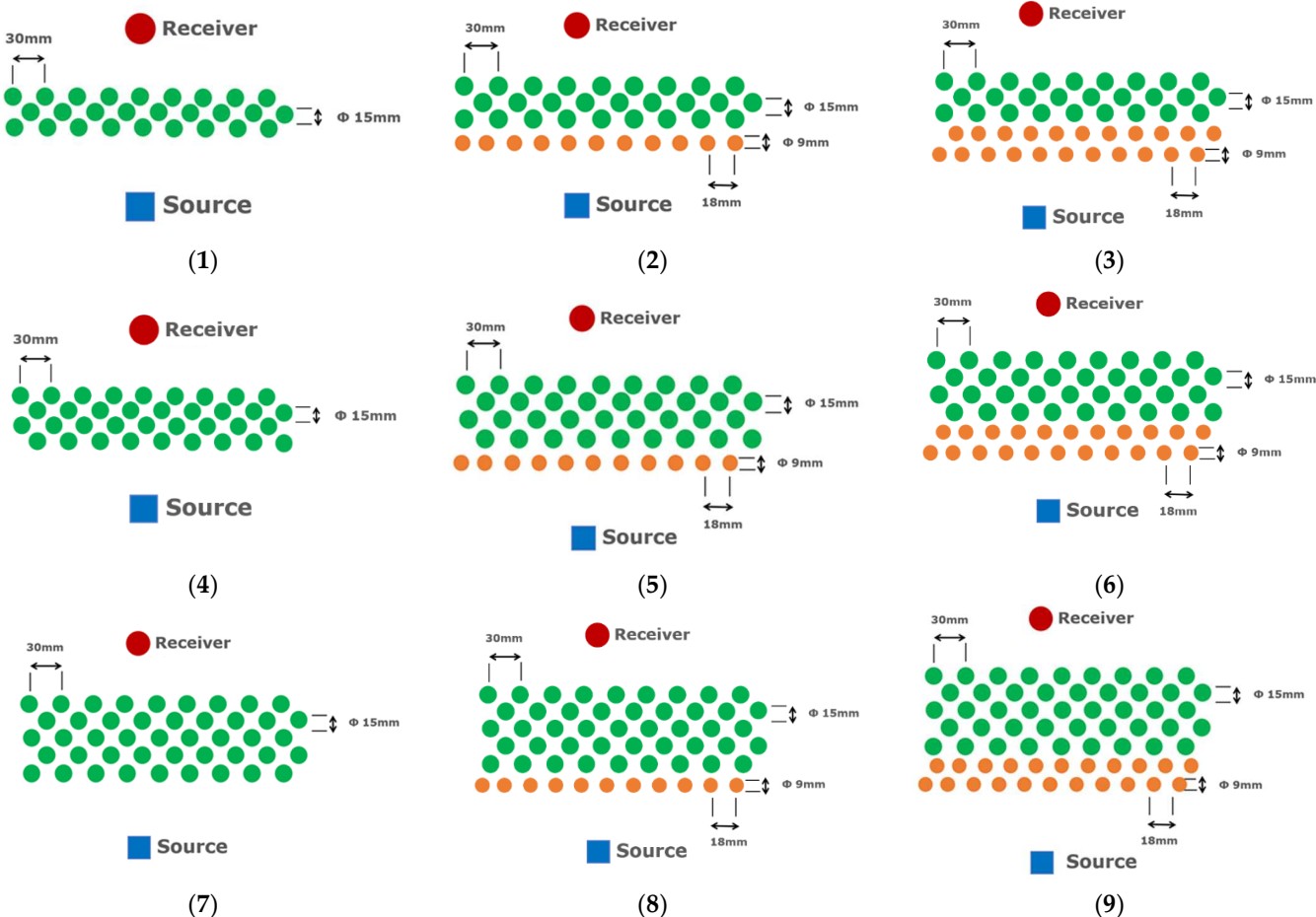

**Figure 5.** Out-of-scale schemes of all the configurations assumed during the acoustic measurements: 3-row type A structure (**1**); 3-row type A and 1-row type B structure (**2**); 3-row type A and 2-row type B structure (**3**); 4-row type A structure (**4**); 4-row type A and 1-row type B structure (**5**); 4-row type A and 2-row type B structure (**6**); 5-row type A structure (**7**); 5-row type A and 1-row type B structure (**8**); 5-row type A and 2-row type B structure (**9**).

The goal of testing different configurations of the metamaterial acoustic barrier is to obtain a robust data output and to assess the sound attenuation by establishing the number of rows attributed to the cylindrical scatterers. A summary of all the configurations is given as follows:

- **Configuration 1**: the metamaterial acoustic barrier is composed of three rows of type A diameter bars, with a staggered bar diameter in order to interrupt the direct line of sight between the source and the receiver. All the other configurations can be considered as variations of this one.
- **Configuration 2**: this setting consists of adding a row of type B diameter scatterers on one side of the barrier.
- **Configuration 3**: the acoustic barrier is composed of three rows of type A diameter bars and two rows of type B diameter sticks installed on the same side.

- **Configuration 4**: the acoustic barrier is composed of four rows of type A diameter bars only.
- **Configuration 5**: the acoustic barrier is composed of four rows of type A diameter sticks and one row of type B diameter bars.
- **Configuration 6**: the acoustic barrier is composed of four rows of type A diameter sticks and two rows of type B diameter bars.
- **Configuration 7**: the acoustic barrier is composed of five rows of type A diameter bars only.
- **Configuration 8**: the acoustic barrier is composed of five rows of type A diameter sticks and one row of type B diameter bars.
- **Configuration 9**: the acoustic barrier is composed of five rows of type A diameter sticks and two rows of type B diameter bars.

Figure 5 schematically shows all the described configurations, indicating in green the cylindrical elements of type A diameter, and in orange the cylindrical elements of type B diameter. Indications of sound source (blue square) and receiver (red circle) positions are also given. The pictures included in Figure 6 shall be intended as schemes.

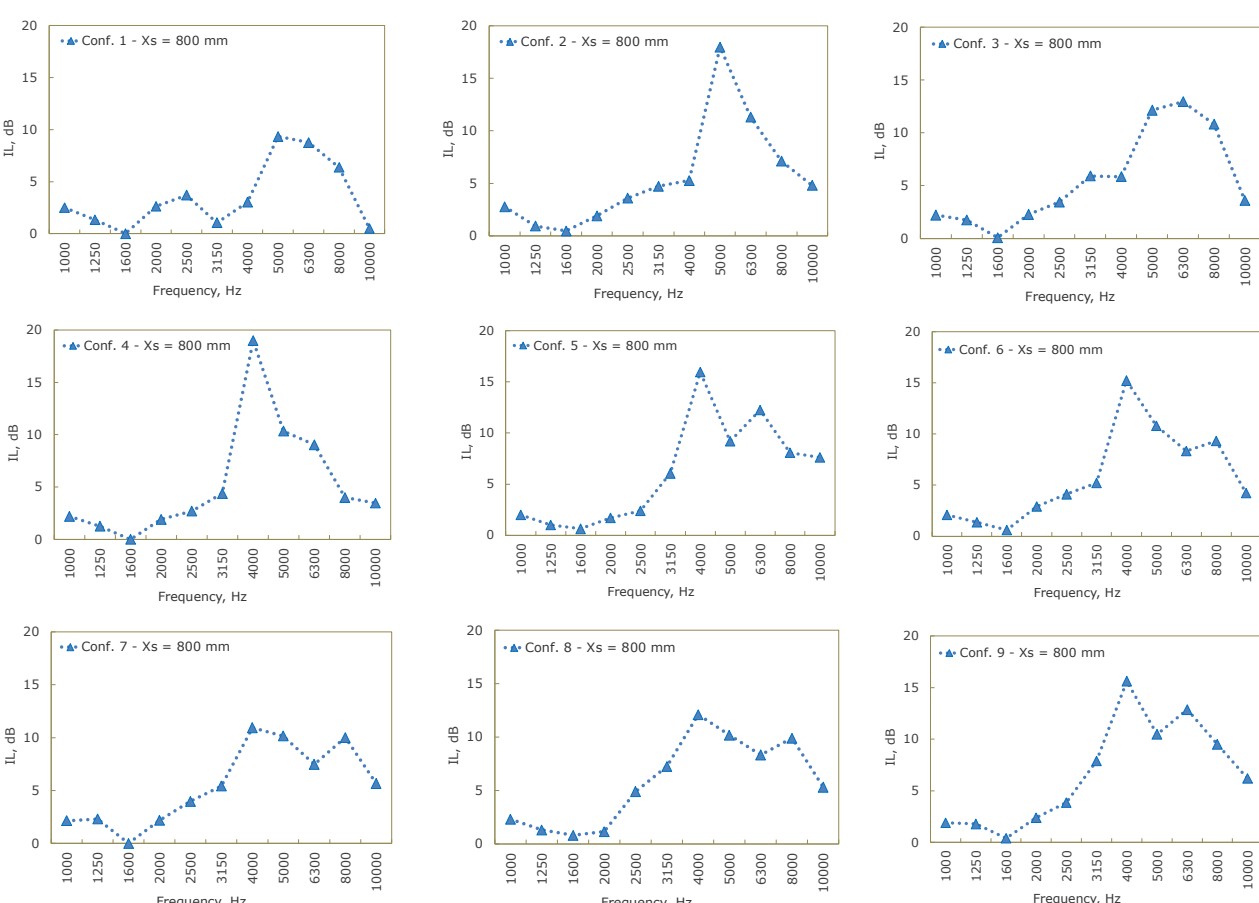

**Figure 6.** Insertional Loss (IL) results related to the first set of measurements. Distance between source and edge of barrier ($X_s$) equal to 800 mm. Distance between receiver and edge of barrier ($X_r$) equal to 800 mm.

Table 1 summarizes the parameters that characterize each configuration.

**Table 1.** Description of all the configurations assumed during the acoustic measurements. D is the scatterer diameter, $\alpha$ is the lattice constant, dr is the distance from the edge of the barrier.

| Configuration | No. of Rows (Type A) | $D_A$ (mm) | No. of Rows (Type B) | $D_B$ (mm) | $\alpha$ (mm) | dr (mm) of Xs—1st Set | dr (mm) of Xs—2nd Set | dr (mm) of Xs—3rd Set |
|---|---|---|---|---|---|---|---|---|
| No. 1 | 3 | 15 | - | - | 30 | | | - |
| No. 2 | 3 | 15 | | | 30 | | | - |
| | | | 1 | 9 | 18 | | | - |
| No. 3 | 3 | 15 | | | 30 | | | - |
| | | | 2 | 9 | 18 | | | - |
| No. 4 | 4 | 15 | - | - | 30 | | | 800 [1] |
| No. 5 | 4 | 15 | | | 30 | | | - |
| | | | 1 | 9 | 18 | 800 | 400 | - |
| No. 6 | 4 | 15 | | | 30 | | | - |
| | | | 2 | 9 | 18 | | | - |
| No. 7 | 5 | 15 | - | - | 30 | | | 800 [1] |
| No. 8 | 5 | 15 | | | 30 | | | - |
| | | | 1 | 9 | 18 | | | - |
| No. 9 | 5 | 15 | | | 30 | | | - |
| | | | 2 | 9 | 18 | | | - |

[1] Receiver moved into 25 positions.

### 2.2.1. Analysis of Multiple Configurations—1st Set of Measurements

The distance between the source and the closest edge of the barrier ($Xs$) during the first set of measurements was 800 mm, equal to the distance existing between the receiver and the barrier ($Xr$). This setting was used for all 9 configurations.

### 2.2.2. Analysis of Multiple Configurations—2nd Set of Measurements

By keeping all the configurations from 1 to 9 as detailed in Section 2.2 and summarized in Figure 6, the second set of measurements involved only the change of distance between the sound source and the edge of the barrier ($Xs$), being reduced to 400 mm. On this basis, all the measurements were repeated by keeping the same characteristics as the 1st set of measurements.

### 2.2.3. Spatial Distribution of the Insertion Loss (IL)—3rd Set of Measurements

The third set of measurements was carried out by following configurations 4 and 6 of the scatterers, composed of four rows of A diameter sticks (conf. 4) and with an additional two rows of B diameter sticks (conf. 6), respectively. The distance between the sound source and the barrier ($Xs$) was kept equal to 800 mm while the microphone was moved into 25 positions, resulting in a grid distance of 200 × 200 mm, as indicated in Figure 4.

## 3. Results

### 3.1. Data Results Related to the 1st Set of Measurements

By analyzing the data in the frequency range between 1.0 kHz and 10 kHz, different behavior in terms of insertion loss (IL) was found. In particular, Figure 7 shows the results related to configurations 1 to 9 of the acoustic barrier, specifically composed of three rows of type A diameter bars and having the addition of one to three rows of sticks of type B diameter.

Figure 6 indicates a peak of IL, equal to 18 dB, in a specific frequency band (i.e., 5 kHz), obtained by adding a row of sticks having type B diameter (configuration 2). This behavior was not found by adding a second row of type B diameter sticks (configuration 3), for which a wider attenuation occurred between 5 kHz and 8 kHz, being around 13 dB and having a dissimilar amplitude.

By having the acoustic barrier composed of four rows of type A diameter sticks (configuration 4), the maximum peak attenuation occurred at 4.0 kHz, equal to 19 dB. By

adding one row of sticks composed of type B diameter, the attenuation was lowered by 3 dB but it became wider, covering the frequencies between 4 kHz and 10 kHz. In configuration 6, the maximum peak attenuation at 4 kHz was slightly lower than configuration 5, equal to 15 dB at 4 kHz.

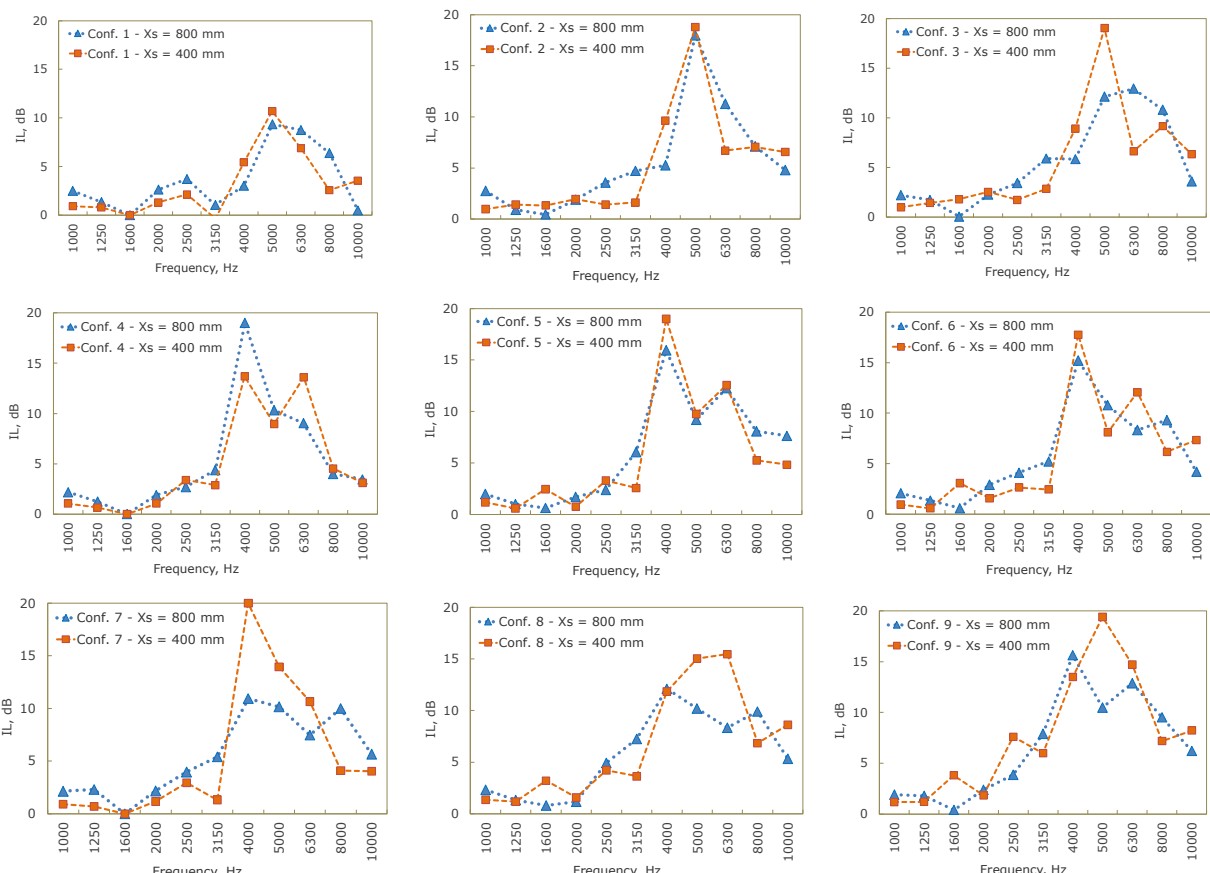

**Figure 7.** Insertional Loss (IL) comparison between 1st and 2nd set of measurements, related to all the configurations of scatterer layout. Distance between source and edge of barrier (*Xs*) is equal to 800 mm for the dotted-blue lines and to 400 mm for the dashed-orange lines. Distance between receiver and edge of barrier (*Xr*) is equal to 800 mm.

When there were 5 rows of bars having type A diameter, a broadband attenuation was recorded at high frequencies (i.e., between 4.0 kHz and 10 kHz). The addition of one row of type B diameter sticks increased the attenuation proportionally to 1 dB, related to the frequency bands falling into the same range. Furthermore, the addition of the second row of type B diameter sticks highlighted a peak attenuation at 4.0 kHz, equal to 16 dB.

Overall, an acoustic barrier composed of three rows of type A diameter elements achieved a sound attenuation of less than 10 dB. The addition of only a row of type B diameter sticks had the effect of pronouncing the IL at the 5.0 kHz frequency band, which was more accentuated with the addition of the second row of type B diameter sticks. The best sound attenuation in terms of IL was recorded with the configurations 5 and 9, equal to 16 dB at 4.0 kHz and having comparable values at the bands between 5.0 kHz and 10 kHz. On this basis, the effectiveness of a coupled sound barrier composed of different sizes of scatterer diameter was demonstrated; therefore, the IL covers a broader frequency band gap, having similar characteristics to vehicular road traffic noise.

### 3.2. Data Results Related to the 2nd Set of Surveys

A similar analysis was undertaken for the data obtained during the second set of measurements. Figure 8 shows the results related to configurations 1 to 9, specifically

composed of three rows of type A diameter sticks and an additional one to three rows of type B diameter sticks. This second set of measurements is characterized by changing the position of the sound source to be 400 mm, instead of 800 mm, from the barrier.

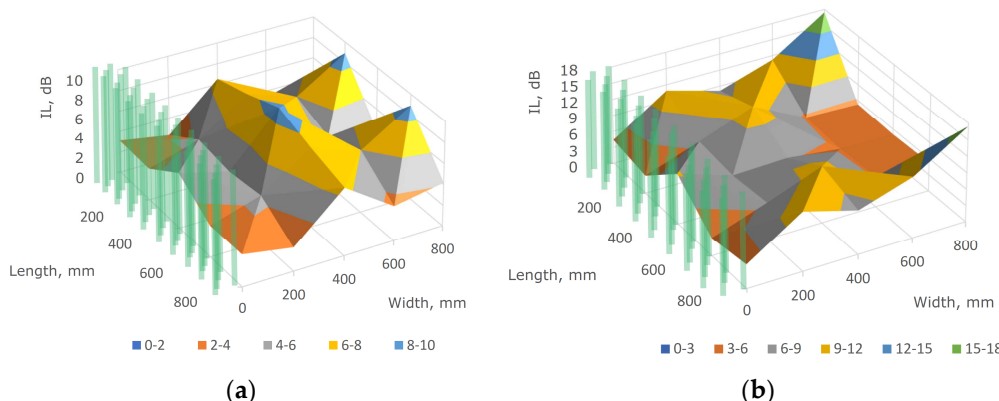

**Figure 8.** 3D view of the spatial distribution related to configuration 4, at 5 kHz (**a**) and 6.3 kHz (**b**). Distance between sound source and edge of barrier ($Xs$) is equal to 800 mm. Distance between receiver and edge of barrier ($Xr$) is variable and corresponds to 25 positions of a 200 × 200 mm grid.

The top-left graph of Figure 8 shows the results related to configuration 1 by placing the sound source at 800 mm and 400 mm from the barrier. The values of the two curves indicate that a similar trend was registered, characterized by spikes pronounced at 5.0 kHz. A similar result is shown in the graph related to configuration 2, where the peaks were found at 5.0 kHz to be very close to each other, around 19 dB. These similarities were not found in the third graph representing configuration 3, where a spike reaching 19 dB was obtained by placing the sound source closer to the barrier only, while an increased distance between the sound source and barrier has the effect of lowering the peak and broadening the attenuation over a wide range of frequency bands (i.e., from 5.0 kHz and 8.0 kHz).

By comparing the results obtained by the first and second sets of measurements, the graph related to configuration 4 indicates a higher IL when the sound source is 800 mm from the barrier, while two peaks at 4 kHz and 6.3 kHz equal to 14 dB were recorded when the sound source is closer to the barrier. This curve trend was not found in configurations 5 and 6, where the spikes at 4 kHz are very similar to the second graph of Figure 7, which indicates a negligible effect of the sound source position from the barrier.

Graphs related to configurations 7, 8, and 9 indicate a clearly efficient performance in terms of sound attenuation when the sound source is placed closer to the barrier. In particular, by having only five rows of type A sticks, IL reaches 20 dB at 4 kHz, almost 10 dB more than when the sound source is placed 800 mm from the barrier. A similar result is obtained in configuration 9, but the frequency recording a high level of IL occurs at 5 kHz. For configuration 8, a broader sound attenuation was found at the frequency range between 4 kHz and 6.3 kHz.

Overall, the cylindrical elements, being the main components of a metamaterial acoustic barrier, determine the sound attenuation in specific frequency bands. The interaction of the soundwave with sticks having different diameters governs the amplitude of the sound attenuation, especially when the sound source is placed closer to the barrier. This means that the values of IL are improved when the distance between the barrier and the sound source is reduced.

### 3.3. Data Results Related to the 3rd Set of Surveys

Additional analysis was undertaken to visualize the spatial distribution of the IL obtained during the third set of measurements. This set of measurements is focused on two types of barriers, related to configuration 4, composed of four rows of type A diameter

sticks, and configuration 6, composed of four rows of type A diameter sticks and two rows of type B diameter sticks.

Based on previous results, a deepening study of the sound source position 800 mm from the barrier was investigated. Acoustic maps were elaborated for the frequency bands equal to 5 kHz and 6.3 kHz, which are the most representative of higher IL values. The acoustic maps were realized with OriginPro2022 software [31], employed to plot the measured results obtained by 25 receiver positions.

Figure 8 shows the IL spatial distribution related to configuration 4, at 5 kHz and 6.3 kHz.

Both graphs of Figure 8 indicate a non-uniform distribution of IL values; IL was found to be lower at a short distance from the barrier and increased at the 400 mm distance, equal to 8 dB. At 6.3 kHz, the greater the distance from the barrier, the less the sound attenuation, although this trend was not recorded at 5 kHz.

Another example of IL spatial distribution is shown in Figure 9, related to configuration 6, at 5 kHz and 6.3 kHz, by adding two rows of type B diameter sticks.

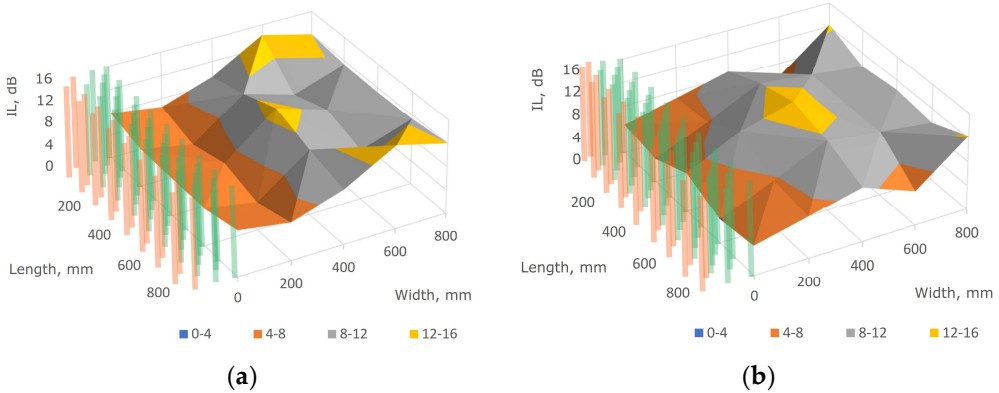

**Figure 9.** 3D view of the spatial distribution related to configuration 6 at 5 kHz (**a**) and 6.3 kHz (**b**). Distance between source and edge of barrier (*Xs*) is equal to 800 mm. Distance between receiver and edge of barrier (*Xr*) is variable and corresponds to 25 positions of a 200 × 200 mm grid.

Figure 9 indicates a better efficiency in terms of IL by adding two rows of type B diameter sticks. In particular, in relation to the first graph, IL values up to 14 dB were found at a 400 mm distance from the barrier values around 8 dB at a 600 mm distance, and even lower at 200 mm from the barrier. Regarding 6.3 kHz, the IL values are more uniform across the grid, at around 10 dB, with a slight peak equal to 13 dB recorded at 400 mm distance from the barrier.

Considering the distance between the source and the edge of barrier (*Xs*) equal to D, this third set of measurements demonstrates that a higher IL was found at D/2, while less attenuation was found very close to the barrier.

## 4. Comparison with Other Research Studies

As anticipated in the introduction, different experiments have been conducted on modular acoustic barriers, carried out on scaled or real-size prototypes. Authors have applied Bragg's law to different forms of metamaterial acoustic barriers; however, the adopted methodologies are varied and comprise preliminary studies conducted with numerical models or acoustic measurements applied to real models (scaled of real size), or both. On this basis, the authors of this paper decided to undertake acoustic measurements of scaled models only.

Although each research study offers an original approach, a comparison can be made with other investigations despite differing methodologies, applications, materials, and analysis. On this basis, some analogies can be found with respect to the studies conducted on the "sonic crystal assisted barrier", as defined by Defrance and Jean [17], composed of

multiple-row scatterers of two different diameters coupled with a conventional opaque barrier and tested with plane-circular or concave rigid scatterers, or even concave scatterers with absorption in the cavity. By considering only the results obtained for the plane-circular scatterers, major sound attenuation was detected around at 1 kHz against the peaks found between 5 kHz and 6300 kHz related to the scaled 1:10 models of this research study, which shall be equivalent to 500 Hz and 630 Hz for real-size dimensions.

Despite the similarities in terms of the band gap, the configuration of the scatterers is different between the reference and the current studies: in the reference research, the barrier is composed of a regular grid of three rows of 0.05 m diameter scatterers, three rows of 0.13 m diameter scatterers, and a plane opaque barrier of 0.2 m thickness. The novelty of this paper introduced by the authors consists of displaying the centers of the circular scatterers to be staggered instead of being installed on a regular grid. The purpose of this methodology is intended to increase the amplitude of the band gap rather than to avoid the installation of any conventional opaque barrier, which instead is intended to be completely replaced.

## 5. Discussions

The performance of acoustic measurements upon the 1:10 scaled models was carried out under controlled conditions, typical of an anechoic chamber, to limit the effects that can potentially affect the results. However, these circumstances are considered more optimistic than real conditions, where the variance of the external gradients is dominant and hence should be taken into account during calculations.

This paper demonstrated that the sound attenuation is variable and dependent on the number of rows composing the barrier, the diameter of the scatterers, and the distance between source and receiver from the barrier.

The measured results highlight that the coupled acoustic barriers create a broader gap band than a barrier composed of only one size of diameter; these outcomes are in line with the principles of mitigation applied to broadband noise, like road traffic.

A short distance between the sound source and the edge of the barrier in all the configurations becomes favorable to increase the IL in the desired frequency bands, ranging between 100 and 1 kHz by reflecting a real scale. In summary, the greater the number of rows composing the barrier, the higher the IL value shifted to high frequencies, emphasized especially when the sound source is at a 400 mm distance from the barrier.

A common factor between the first and second sets of measurements, independently from the specific configuration, is that the sound attenuation was found mainly at 4–5 kHz, corresponding to the spikes reaching up to 19 dB. Based on the third set of measurements, the distance of the receiver from the barrier is also a variable to be considered during the design of an acoustic barrier. By considering D as the distance between the source and the barrier, the results obtained by the third set of measurements indicate that the IL values are higher when the receiver is at D/2 from the barrier, in configurations 4 and 6, at 5 kHz and 6.3 kHz. These values are found to be reduced when the receiver is at a shorter and further distance from the barrier.

Since each environmental site is characterized by peculiarities, it would be difficult to make a standard barrier for all the outdoor contexts. As such, the height of the barrier plays an important role. Although this research study is not focused on the height variance of the barrier, we found that a metamaterial acoustic barrier having a maximum thickness of 0.80 m and height of 3.0 m (as real dimensions and scaled from a 1:10 model) can be applied along the external edges of boulevards and wide roads, as infrastructure of suburban areas characterized by a flow rating around 70–80 km/h (defined as average-speed limit), where the side properties are no more than two-story buildings. This range can be considered restricted for big cities where the blocks are dominated by skyscrapers and tall buildings; however, it can represent an applicable solution for many villages and small towns where the buildings are most likely to be less than two stories.

In terms of structural stability, a multiple-row acoustic barrier does not require the deep foundations of opaque assemblies, which are more sensitive to wind direction and wind speed. Regarding maintenance, the dust that can accumulate in the gaps between scatterers at grade level, while not visually pleasant, does not limit the efficiency.

## 6. Conclusions

This paper deals with the study of sound attenuation expressed in terms of insertion loss (IL) related to different configurations of metamaterial acoustic barriers. A brief introduction is given in relation to the use of modular elements (composed of solid circled scatterers and lattice constants) employed in the field of applied acoustics and noise control. The combination of two diameters chosen for the wooden scatterers (installed for staggered diameter size) and the placement of the sound source in two positions (at 400 mm and 800 mm from the edge of the barrier) allow for a robust data analysis. The sound attenuation gap band was found between 4 kHz and 12.5 kHz, dependent on the number of rows installed to compose the acoustic barrier [32]. A common factor among the results is that the greater the number of rows composing the barrier, the higher the IL value.

The benefit of having an acoustic barrier made of modular elements instead of an opaque surface area could represent an alternative solution to mitigate vehicular road traffic noise, without the necessity of building deep foundations. This was assessed to be applicable, especially along the sides of boulevards and wide roads, where the average speed limit is around 70–80 km/h, in the context of small towns and villages with the prevalence of two-story buildings [33]. The efficiency of a coupled acoustic barrier was found in a broader bandwidth of sound gap attenuation, reflecting the characteristics of road traffic noise.

Further research studies will be focused on the combination of different sizes of lattice constants, as well as on different sizes of scatterer diameters. This methodology aims to enlarge the band gap, especially at low frequencies. However, it was demonstrated that the thinner diameter assigned to scatterers shifts the sound attenuation towards higher frequency bands, in accordance with Bragg's theory. A larger size of scatterer diameter should be considered as more effective at low frequencies in real dimensions, which can be represented by trees.

One of the potential applications of this study could be extended also to the filters installed for attenuating the steady noise of heating, ventilation and air conditioning (HVAC) systems [34], where the noise generated by the fan is centered upon a specific octave band (pure tone). For this scope, the model size tested would be suitable for the cabins and enclosures of the mechanical units. The application of the multiple-row acoustic barriers composed of modular elements, could find development in the field of mechanical ventilation.

**Author Contributions:** Conceptualization, G.I. and A.T.; methodology, G.I.; software, I.L.; validation, A.B.; formal analysis, A.B.; investigation, G.I.; resources, A.T.; data curation, A.B. and I.L; writing—original draft preparation, G.I. and I.L.; writing—review and editing, A.B and A.T.; visualization, A.B.; supervision, G.I.; project administration, A.T.; funding acquisition, A.T. All authors have read and agreed to the published version of the manuscript.

**Funding:** This work was financially supported by Ministero dell'Istruzione, dell' Università e della Ricerca (MIUR)-PRIN 2017 Progetto Settore PE8 Codice 2017T8SBH9_001; "Theoretical modeling and experimental characterization of sustainable porous materials and acoustic metamaterials for noise control".

**Conflicts of Interest:** The authors declare no conflict of interest.

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
