# Peer review of "2D Sonic Acoustic Barrier Composed of Multiple-Row Cylindrical Scatterers: Analysis with 1:10 Scaled Wooden Models"

_applsci, doi:10.3390/app12136302_

Round 1
Reviewer 1 Report
This is a review of applsci-1707211, titled: "Metamaterials acoustic barrier". The authors describe an empirical experiment of a sonic-crystal-based noise barrier. There are numerous concerns that render this submission not suitable to be published as an original research article. The authors may want to condense the experimental results to publish this as a short communication instead. If the authors wish to resubmit, the following concerns must be addressed.
- The title is too generic. What is the key contribution here?
- The introduction adds nothing to the field. Sonic crystals have been well documented, there's even a review article exactly on sonic crystal noise barriers that was not even cited https://www.mdpi.com/2076-3298/6/2/14/htm
- The research question is missing. What problem specific problem are you trying to solve in the field of sonic crystal noise barriers? What is the benefit of wooden bars over other types? The introduction should address all the recent research related to the research question.
- The structure of the paper is misleading. Materials and methodologies should include the experimental method and materials used but instead went to section 3 as a separate section?
- The formula for insertion loss must be included
- Figure 5 and the description of the configurations in 3.1 should be organized in a full-page table with all the relevant parameters neatly labeled in categories.
- The goal of the configurations is not clear. What are the authors trying to find out by having different layouts? This points back to the missing research question.
- "acoustic maps have been realized with a software" -> what software? it is not possible to replicate your experiments with missing information on the tools
- Authors should explain why numerical modeling is not used to do iterative design before experimental validation?
- What are the limitations of sonic crystal barriers? Does it still have the same problem of being limited to the "shadow zone"? What about noise propagating over the barrier to high-rise buildings?
Author Response
This is a review of applsci-1707211, titled: "Metamaterials acoustic barrier". The authors describe an empirical experiment of a sonic-crystal-based noise barrier. There are numerous concerns that render this submission not suitable to be published as an original research article. The authors may want to condense the experimental results to publish this as a short communication instead. If the authors wish to resubmit, the following concerns must be addressed.
- The title is too generic. What is the key contribution here?
Dear Reviewer, the title has been changed accordingly
- The introduction adds nothing to the field. Sonic crystals have been well documented, there's even a review article exactly on sonic crystal noise barriers that was not even cited https://www.mdpi.com/2076-3298/6/2/14/htm
Dear Reviewer, the introduction has been changed with the integration taken from the suggested publication.
- The research question is missing. What problem specific problem are you trying to solve in the field of sonic crystal noise barriers? What is the benefit of wooden bars over other types? The introduction should address all the recent research related to the research question.
Dear Reviewer, the text has been reworded including the suggested topics and by explaining the choice of our selection, in terms of material and shape of the elements composing the barrier.
- The structure of the paper is misleading. Materials and methodologies should include the experimental method and materials used but instead went to section 3 as a separate section?
Dear Reviewer, section 2 and 3 have been combined
- The formula for insertion loss must be included
Dear Reviewer, it has been included
- Figure 5 and the description of the configurations in 3.1 should be organized in a full-page table with all the relevant parameters neatly labeled in categories.
Dear Reviewer, a table summarizing all the parameters for each configuration has been added.
- The goal of the configurations is not clear. What are the authors trying to find out by having different layouts? This points back to the missing research question.
Dear Reviewer, the authors have explained the implementation of the research in the field of metamaterials acoustic barriers for the mitigation of road traffic noise by doing tests on coupled rows of different size diameter of scatterers.
- "acoustic maps have been realized with a software" -> what software? it is not possible to replicate your experiments with missing information on the tools
Dear Reviewer, thank you for the observation. Name of the software has been added
- Authors should explain why numerical modeling is not used to do iterative design before experimental validation?
Dear Reviewer, some research studies of sonic crystals are focused only onto predictions, some onto measurements, and others on both. The authors preferred to increase the databank obtained by measurements. It has been explained.
- What are the limitations of sonic crystal barriers? Does it still have the same problem of being limited to the "shadow zone"? What about noise propagating over the barrier to high-rise buildings?
Dear Reviewer, comments on this topic have been added
Reviewer 2 Report
This paper deals with the study of the acoustic insulation performance of post sandwich systems used as sound insulation barriers of low frequency noise in the range from 100 Hz to 1 kHz.
The subject matter of the article is interesting, however, the way of research presentation and the adopted research model in the reviewer's opinion do not allow its publication in Applied Sciences.
Selected remarks are presented below:
1. The last two paragraphs of the introduction provide a summary of the methodology and research results. This information should be moved to the chapters it applies to. In the reviewer's opinion, information about the shortcomings of the subject matter addressed in the article and an indication of the novelty of the research undertaken by the authors should be indicated in the introduction.
2. The materials and methods chapter basically provides information in terms of the literature review. It shouldn't be here.
3. The Acoustic Measurements and Equipment chapter contains information on materials and test methods.
4. In my opinion, the adopted research model leading to dimension scale and attenuation frequency conversions should be rigorously described theoretically and confirmed by a comparative study.
5. Data on the characteristics of the materials used (stiffness, density, ...) are missing.
6. Figure number 1 is unclear. It is advisable to show the direction of wave propagation and the points and types of wave scattering. Further drawings are very large, sometimes taking up more than two typewritten pages while presenting a small range of information.
7. The discussion section should include a comparison of the results obtained with studies conducted by other researchers that were presented in the literature review.
The information presented in this chapter is a description of the research results obtained and should be there.
8. The conclusion section includes a summary of the article and a literature survey. Conclusions should be bulleted.
Yours sincerely,
Reviewer.
Author Response
This paper deals with the study of the acoustic insulation performance of post sandwich systems used as sound insulation barriers of low frequency noise in the range from 100 Hz to 1 kHz.
The subject matter of the article is interesting, however, the way of research presentation and the adopted research model in the reviewer's opinion do not allow its publication in Applied Sciences.
Selected remarks are presented below:
1. The last two paragraphs of the introduction provide a summary of the methodology and research results. This information should be moved to the chapters it applies to. In the reviewer's opinion, information about the shortcomings of the subject matter addressed in the article and an indication of the novelty of the research undertaken by the authors should be indicated in the introduction.
Dear Reviewer, the introduction has been reworded by reflecting your observations as well.
- The materials and methods chapter basically provides information in terms of the literature review. It shouldn't be here.
Dear Reviewer, general considerations have been moved to the introduction.
- The Acoustic Measurements and Equipment chapter contains information on materials and test methods.
Dear Reviewer, paragraphs related to methods have been moved to section 2
- In my opinion, the adopted research model leading to dimension scale and attenuation frequency conversions should be rigorously described theoretically and confirmed by a comparative study.
Dear Reviewer, this observations have been accomplished
- Data on the characteristics of the materials used (stiffness, density, ...) are missing.
Dear Reviewer, the requested data have been added
- Figure number 1 is unclear. It is advisable to show the direction of wave propagation and the points and types of wave scattering. Further drawings are very large, sometimes taking up more than two typewritten pages while presenting a small range of information.
Dear Reviewer, Figure 1 has been changed and the other drawings have been organized differently, as per your suggestion.
- The discussion section should include a comparison of the results obtained with studies conducted by other researchers that were presented in the literature review.
The information presented in this chapter is a description of the research results obtained and should be there.
Dear Reviewer, comparison with other research studies has been made.
- The conclusion section includes a summary of the article and a literature survey. Conclusions should be bulleted.
Dear Reviewer, based on the rewording, bullets are not necessary and do not apply here. Sorry
Reviewer 3 Report
Overall the paper is well-written, interesting, and provides a basic acoustic analysis of the performance of some cylindrical-based acoustic meta-materials. My major comment is that I strongly recommend revising the presentation of the results by using non-dimensionalized parameters, to aid the performance comparison of each configuration. Also, Figs 13-14 must be checked.
1 - Is it possible to use non-dimensionalized parameters for Phi and other distances, instead of mm ? Can some kind of 'porosity' parameter also be used ? (eg - the ratio of the area of the circles, divided by the rectangular area in which they exist). This will make your work easier to compare with other studies and more likely be referenced by future works.
2 - Can the information of the configs in Section 3.1 be condensed into a table ? Also, put the DAQ information in Section 3, not 3.1, as it is relevant for every test.
3 - Please include an insertion loss (IL) equation in the paper, around Page 4 lines 129-131
4 - I recommend making Figure 6 either a 3 x 3 subplot or a 4 (down) x 2 (wide) subplot to improve its appearance in the paper
5 - In Fig 6, some configurations are thicker than others - so where exactly is Xs and Xr? In Fig. 4, the location at which Xs ends and Xr starts is not clear - I assume that in all tests, regardless of the thickness of the barrier, this "origin" is the same? This origin and its location must be made clear, and again, I recommend some form of non-dimensionalization.
6 - A thicker material will likely have better acoustic absorption - so comparing config. L with config H, for example, naturally there is a greater IL - can you normalize IL with overall thickness? Then you can see how effective that material configuration really is. Otherwise you'll won't know if the improved absorption is due to thickness, config design, or both.
7 - I recommend removing the bar-type plots in Fig 9 and replace them with line-plots like in Fig 10 , for consistency.
8 - Figs 7,8,9 should be made into one large 3 x 3 subplot and similarly for Figs 10,11,12.This will help the overall paper layout and readability.
9 - Figs 13,14 must include some spatial axes. Also, why does the data appear discontinuous across the vertical direction ? Have separate data regions been processed separately, and then pasted together after (maybe)? This must be corrected.
10 - Please add more discussion comparing the performance of these meta materials with other acoustic absorbing materials of similar thicknesses, testing conditions, etc. While meta materials produce good IL, many other materials do - how do meta materials compare against them ?
11 - For most figs - please improve the captions and explain what each figure is in more detail
12 - configs go from "H" to "L" ... any reason why "I, J, K" were skipped ?
13 - I recommend adding another figure like Fig 13, but using overall IL (calculated by integrating the IL spectrum) - this would be more useful overall to the reader to show how well the meta materials reduce noise, overall, as compared to some specific frequencies.
Author Response
Overall the paper is well-written, interesting, and provides a basic acoustic analysis of the performance of some cylindrical-based acoustic meta-materials. My major comment is that I strongly recommend revising the presentation of the results by using non-dimensionalized parameters, to aid the performance comparison of each configuration. Also, Figs 13-14 must be checked.
1 - Is it possible to use non-dimensionalized parameters for Phi and other distances, instead of mm ? Can some kind of 'porosity' parameter also be used ? (eg - the ratio of the area of the circles, divided by the rectangular area in which they exist). This will make your work easier to compare with other studies and more likely be referenced by future works.
2 - Can the information of the configs in Section 3.1 be condensed into a table ? Also, put the DAQ information in Section 3, not 3.1, as it is relevant for every test.
Dear Reviewer, this has been done
3 - Please include an insertion loss (IL) equation in the paper, around Page 4 lines 129-131
Dear Reviewer, this has been done
4 - I recommend making Figure 6 either a 3 x 3 subplot or a 4 (down) x 2 (wide) subplot to improve its appearance in the paper
Dear Reviewer, this has been done accordingly
5 - In Fig 6, some configurations are thicker than others - so where exactly is Xs and Xr? In Fig. 4, the location at which Xs ends and Xr starts is not clear - I assume that in all tests, regardless of the thickness of the barrier, this "origin" is the same? This origin and its location must be made clear, and again, I recommend some form of non-dimensionalization.
Dear Reviewer, Xs and Xr have been specified to be from the edge of the barrier to source o receiver. About the form of non-dimensionalization, this has been done for the type of diameter of the scatterers. It has been considered that doing the same for the distance of the source and receiver from the barrier, it may create confusion although it is clear that the experiments have been done by halving the distance from the fist set of measurements.
6 - A thicker material will likely have better acoustic absorption - so comparing config. L with config H, for example, naturally there is a greater IL - can you normalize IL with overall thickness? Then you can see how effective that material configuration really is. Otherwise you'll won't know if the improved absorption is due to thickness, config design, or both.
7 - I recommend removing the bar-type plots in Fig 9 and replace them with line-plots like in Fig 10 , for consistency.
Dear Reviewer, bar graphs have been substituted accordingly
8 - Figs 7,8,9 should be made into one large 3 x 3 subplot and similarly for Figs 10,11,12.This will help the overall paper layout and readability.
Dear Reviewer, the figures grouped into a 3x3 table.
9 - Figs 13,14 must include some spatial axes. Also, why does the data appear discontinuous across the vertical direction ? Have separate data regions been processed separately, and then pasted together after (maybe)? This must be corrected.
Dear Reviewer, the graphs of Figs. 13 and 14 have been done in 3D
10 - Please add more discussion comparing the performance of these meta materials with other acoustic absorbing materials of similar thicknesses, testing conditions, etc. While meta materials produce good IL, many other materials do - how do meta materials compare against them?
Dear Reviewer, comparisons with other tests have been made
11 - For most figs - please improve the captions and explain what each figure is in more detail
Dear Reviewer, captions have been improved
12 - configs go from "H" to "L" ... any reason why "I, J, K" were skipped ?
Dear Reviewer, name of configurations has been changed to numerical order.
13 - I recommend adding another figure like Fig 13, but using overall IL (calculated by integrating the IL spectrum) - this would be more useful overall to the reader to show how well the meta materials reduce noise, overall, as compared to some specific frequencies.
Dear Reviewer, the graphs of Figs. 13 and 14 have been done in 3D, as per above
Round 2
Reviewer 1 Report
Although the authors have partially satisfied the comments raised in the previous review, there are still major concerns as follows:
This work appears to extend from [34] and should be mentioned as such. The original title and flow of the article are exactly the same as [34] and constitute an attempt at self-plagiarism. The extension from [34] to multiple rows does not seem to be sufficient for a full article and should be written as a short communication instead.
Title: If wood is the key idea discussed then it should be in the title. There are also grammatical errors in the title, i.e. metamaterials, multiple-rows.
Introduction: As mentioned previously, there are many parts that have been discussed extensively in other papers that you can refer the readers to. This paper should extend from [34] and not just paraphrase what is written in [34]. Moreover, equation (1) is not referenced at all in the rest of the paper and hence it does not seem necessary. The authors should focus and expand on why wood and why multi-row configurations? Also, there is an inconsistency in that bamboo was mentioned in the introduction but in the end, the material used was wood. I believe that bamboo is more sustainable (being one of the fastest growing plants) and durable (more fire resistant) than wood? Lastly, it does not seem prudent to just "comply" with other research studies i.e. [19,20], which are from less reputable sources. I would suggest looking at Fig. 9 in "[1] U. Sandberg, Tyre/ road noise - Myths and realities, in: INTERNOISE NOISE-CON Congr. Conf. Proc., Institute of Noise Control Engineering, The Hague, The Netherlands, 2001: pp. 2608–2629. https://www.ingentaconnect.com/content/ince/incecp/2001/00002001/00000002/art00034." for where most of the energy of road traffic noise resides.
Materials and methods: It seems that the authors are not familiar with what should be in the materials and methods section. The discussion on why wood was chosen as the material to investigate should be in the introduction. This section should contain the experiment procedure, stimuli, experiment setup, etc. That means sections 2 and 3 should be combined under "materials and methods" with appropriate sub-sections.
Equipment: The single loudspeaker in the experiment appears to be a tweeter, which puts the validity of the experiment in question. The authors should explain how a point source speaker can be used to reproduce the wavefront of traffic noise, which are moving point sources? It also appears that the speaker is pointing toward the ground? There should be a plot to show the free-field response of the speaker as it does not seem possible to be flat response as claimed.
Results: Figure captions should be self-contained. There should be a rather detailed description of the figure and not just a simple statement.
Author Response
Although the authors have partially satisfied the comments raised in the previous review, there are still major concerns as follows:
This work appears to extend from [34] and should be mentioned as such. The original title and flow of the article are exactly the same as [34] and constitute an attempt at self-plagiarism. The extension from [34] to multiple rows does not seem to be sufficient for a full article and should be written as a short communication instead.
Dear Reviewer, the title has been revised and a clear statemen that this article is a development of the previous research has been added into the introduction.
Title: If wood is the key idea discussed then it should be in the title. There are also grammatical errors in the title, i.e. metamaterials, multiple-rows.
Dear Reviewer, the title has been changed accordingly.
Introduction: As mentioned previously, there are many parts that have been discussed extensively in other papers that you can refer the readers to. This paper should extend from [34] and not just paraphrase what is written in [34].
Dear Reviewer, comments as per above
Moreover, equation (1) is not referenced at all in the rest of the paper and hence it does not seem necessary.
Dear Reviewer, equation (1) has been requested by another reviewer
The authors should focus and expand on why wood and why multi-row configurations? Also, there is an inconsistency in that bamboo was mentioned in the introduction but in the end, the material used was wood. I believe that bamboo is more sustainable (being one of the fastest growing plants) and durable (more fire resistant) than wood?
Dear Reviewer, choice of plain wood vs bamboos has been explained.
Lastly, it does not seem prudent to just "comply" with other research studies i.e. [19,20], which are from less reputable sources. I would suggest looking at Fig. 9 in "[1] U. Sandberg, Tyre/ road noise - Myths and realities, in: INTERNOISE NOISE-CON Congr. Conf. Proc., Institute of Noise Control Engineering, The Hague, The Netherlands, 2001: pp. 2608–2629. https://www.ingentaconnect.com/content/ince/incecp/2001/00002001/00000002/art00034." for where most of the energy of road traffic noise resides.
Dear Reviewer, this reference has been added
Materials and methods: It seems that the authors are not familiar with what should be in the materials and methods section. The discussion on why wood was chosen as the material to investigate should be in the introduction. This section should contain the experiment procedure, stimuli, experiment setup, etc. That means sections 2 and 3 should be combined under "materials and methods" with appropriate sub-sections.
Dear Reviewer, modifications to the sections have been made
Equipment: The single loudspeaker in the experiment appears to be a tweeter, which puts the validity of the experiment in question. The authors should explain how a point source speaker can be used to reproduce the wavefront of traffic noise, which are moving point sources? It also appears that the speaker is pointing toward the ground? There should be a plot to show the free-field response of the speaker as it does not seem possible to be flat response as claimed.
Dear Reviewer, the authors have explained the limits of these experiments and also described the way to optimize the tests by moving the receiver (instead of the sound source) to create the directivity angles.
Results: Figure captions should be self-contained. There should be a rather detailed description of the figure and not just a simple statement.
Dear Reviewer, all the figure captions have been revised
Reviewer 2 Report
The article has been substantially improved from the previous version.
In my opinion, the following should be corrected before publication:
- the introduction should demonstrate the gaps in existing knowledge in the topic addressed and the reasons why the research was undertaken.
- the novelties and their contribution to the field of the addressed subject of knowledge should be rigorously demonstrated in the results and discussion of the article.
- The manner in which conclusions concerning the properties of full-size (3 m. height) partitions are derived from the presented scale tests should be rigorously described.
Yours Sincerely,
Reviewer.
Author Response
The article has been substantially improved from the previous version.
Dear Reviewer, thank you so much
In my opinion, the following should be corrected before publication:
- the introduction should demonstrate the gaps in existing knowledge in the topic addressed and the reasons why the research was undertaken.
Dear Reviewer, this has been addressed
- the novelties and their contribution to the field of the addressed subject of knowledge should be rigorously demonstrated in the results and discussion of the article.
Dear Reviewer, the novelty of this study consists of the test undertaken on a varied methodologies combined together: coupled barrier by using different size of scatterer’s diameter, disposition of the scatterers staggered with respect to a perpendicular grid. In addition, the tests carried out on different distance between source and barrier have been compared with other research studies.
- The manner in which conclusions concerning the properties of full-size (3 m. height) partitions are derived from the presented scale tests should be rigorously described.
Dear Reviewer, this observation should be self-explanatory since the tested model is 1:10 scaled. A dimension of 300mm would be in real dimensions equal to 3000mm, equal to 3m. However, it was already contextualized that this solution can be applicable to wide boulevards of small towns where the average height of properties is about 2-storey, typical of suburban areas. In addition, it has been introduced the potentiality of this model to be used like a filter for the mechanical ventilation units. In this specific case, the dimensions of the model been tested shall be suitable for the size of cabin/enclosures of the units, but it will be deeply studied for future publications.
Round 3
Reviewer 1 Report
Although I appreciate your effort, it was quite disappointing that I had to point out potential self-plagiarism.
Nevertheless, I have one last suggestion to fix the awkwardness in the title -> "2D Sonic acoustic barrier composed of multiple-row cylindrical scatterers: an analysis with 1:10 scaled wooden models"
Lastly, there is still a need for English editing as indicated in the options in my previous reviews.
Author Response
Dear Reviewer,
Thank you for your comments which have allowed us to improve the research. The paper is the continuation of the one published in the Applied Acoustics journal (Iannace, G.,. Et al Metamaterials acoustic barrier. Applied Acoustics, 2021, 181, 108172). In this paper we have measured the effects of parallel rows with different diameters because our aim is to apply the barriers to real cases and the scale model provides indications on how to proceed in the full scale condition. We have noticed that there are no measures on this topic.
Thank you.
Reviewer 2 Report
All my comments have been corrected and/or explained. In my opinion, the article should be accepted in its present form.
Yours Sincerely,
Reviewer.
Author Response
we have revised the paper